# Experience of the COVID-19 Pandemic in the Care of Patients with Predominantly Antibody Deficiencies (PADs)—A Qualitative Study with Perspectives from Both Patients and Nurses

**DOI:** 10.3390/nursrep15030104

**Published:** 2025-03-18

**Authors:** Ramona Fust, Sofia Nyström, Britt Åkerlind, Åsa Nilsdotter-Augustinsson, Christina Petersson

**Affiliations:** 1Department of Infectious Diseases in Region Östergötland, Linköping University, 583 30 Linköping, Sweden; asa.nilsdotter-augustinsson@liu.se; 2Department of Biomedical and Clinical Sciences, Linköping University, 583 30 Linköping, Sweden; sofia.c.nystrom@liu.se; 3Department Clinical Immunology and Transfusion Medicine, Linköping University, 583 30 Linköping, Sweden; 4Department of Infection Control and Hygiene, Linköping University Hospital Sweden, 583 30 Linköping, Sweden; britt.akerlind@regionostergotland.se; 5Jönköping Academy for Improvement of Health and Welfare, Jönköping University, 551 11 Jönköping, Sweden; christina.petersson@ju.se

**Keywords:** telemedicine, coordination of care, everyday life, chronic conditions, nursing

## Abstract

**Background/Objectives:** One of the risk groups during the COVID-19 pandemic was people with predominantly antibody deficiencies (PADs) that have a compromised immune system. In the absence of evidence and clinical experience, there were challenges for patients in their daily life and for staff in counseling during this time. Therefore, the aim of this study was to explore the experiences of PAD patients and nurses during the COVID-19 pandemic. **Methods:** Focus group interviews with patients (n = 12) and nurses (n = 12) were performed separately, which were then analyzed using content analysis. **Results:** The daily life of PAD patients was affected during the pandemic, with concerns about becoming seriously ill. Social isolation and adherence to recommendations by the majority of the Swedish population resulted in patients feeling infectiously healthier during this period. The rapid transition of specialist care to telemedicine care encounters was an important measure taken to address patients’ concerns and questions according to both patients and nurses. In addition, patients expressed a need for a coordinated care plan to facilitate access to integrated care. **Conclusions:** The high level of trust for authorities in Sweden was related to the high compliance with the recommendations, which reduced the spread of the infection. The role of specialized care is an important support for PAD patients, which was particularly evident during the pandemic. Information transfer to a specific risk group, such as people with PADs, is important and can usefully be coordinated by their specialist clinic. Telemedicine meetings are an important complement for people with PADs and need to be further elaborated. Also, there is a need to clarify how to better coordinate primary and specialized care.

## 1. Introduction

In January 2020, cases of the new coronavirus were reported in Wuhan, China. The coronavirus became known as SARS-CoV-2. The disease was named COVID-19 [1]. On 11 March 2020, WHO declared COVID-19 a pandemic. In some individuals, COVID-19 develops into a severe life-threatening acute respiratory syndrome [2]. The Public Health Agency of Sweden (PHAS) is an independent public authority under the Swedish Government, responsible for the overall national protection of the population against communicable diseases and the coordination of infection control at a national level. This authority produces regulations, recommendations and guidance for healthcare professionals to ensure effective infection control. Sweden’s COVID-19 mitigation strategy differed from the rest of the world (Appendix A). It was rather based on voluntary actions by citizens compared to many other countries that introduced legal measures to control the outbreak of COVID-19 [2]. In Sweden, there are 21 administrative counties or regions, and each region has its own physician responsible for Communicable Disease Prevention and Infection Control. This physician issues regional and local regulations and recommendations, and these may differ between regions [1]. Like the general population, people with predominantly antibody deficiencies (PADs) and their families had to rely on PHAS guidelines. During the first year of the COVID-19 pandemic, the media daily reported on the rapid spread of the SARS-CoV-2 virus and the risk of serious illness, especially those in various risk groups and people with inborn errors of immunity (IEI) [3].

### 1.1. Care of PAD Patients

Globally, many patients with IEI, such as PADs, remain undiagnosed and neither the prevalence nor incidence are certain for these conditions [4]. The largest IEI group is characterized by impaired humoral immune responses and a lack of antibodies [5]. Today, there are more than 430 genetically defined conditions associated with impaired immune responses [6]. However, in most cases, the underlying cause of PAD is unknown, and PAD is a diagnosis of exclusion. There are around 45 known gene variants associated with PAD causing B-cell failure and hypogammaglobinemia, and about 35% of patients with PAD carry disease causing gene variants [5]. In Sweden, the estimated prevalence of IEI is 40,000 people, and 500 of those are living with severe PAD, including Common Variable Immunodeficiency (CVID) and X-linked agammaglobulinemic (XLA). People affected by severe PAD are characterized by impaired immune functions and poor antibody responses. They are more susceptible to respiratory tract infections that can be life-threatening and chronic [7]. Immunoglobulin (Ig) replacement therapy and antibiotics are important to control infections [4,8]. In addition to increased infections, a significant proportion of people affected by severe PAD also suffer from immune dysregulation, which is associated with increased morbidity [9]. People with PADs report lower perceived health-related quality of life (HRQoL) compared to the general population or people with other chronic diseases, such as diabetes mellitus or juvenile idiopathic arthritis [10]. Immunoglobulin (Ig) replacement therapy is central to the treatment of people living with PAD, reducing the number of infections and having a positive impact on HRQoL [11,12]. Ig replacement therapy is administered as subcutaneous or intravenous infusions on an outpatient basis or as home treatment for those with self-administered subcutaneous infusions [13]. The need to take Ig replacement therapy can be perceived as a burden of treatment since time must be allocated for treatments either in hospital or at home [12].

In Sweden, care for people with severe PAD is provided by the staff at a center for immunodeficiency diseases, usually organized by a department of infectious diseases at a secondary or tertiary hospital. Sweden has a long tradition of self-administered subcutaneous Ig replacement therapy [14]. About 91.5% of people with PAD administer Ig replacement treatment at home [15]. Nurses involved in the care of people with PAD play an important coordinating role, providing physical, emotional, and psychosocial support with the aim of preventing ill health in this vulnerable group of patients [16].

### 1.2. Rationale of the Study

People with PAD were considered a risk group for COVID-19. Although there was no evidence at the beginning of the pandemic on how they would be affected by the virus, they were still considered a risk group vulnerable to develop severe COVID-19 [17]. Also, it was uncertain whether people with PAD would develop protective anti-SARS-CoV-2 immunity after vaccination [18]. The experience of people with PAD may provide knowledge on how to advise and support patients with rare diseases in their daily lives during a public health crisis. The aim of the present study was to explore the experiences of how the COVID-19 pandemic affected people with PAD, based on interviews with patients and nurses specially trained to care for them.

## 2. Materials and Methods

This study is part of a larger research project on patients with severe immunodeficiency and used a descriptive design based on a qualitative method [19]. Participants were included from five different immunodeficiency outpatient clinics in southern Sweden. There are a total of 15 immunodeficiency registered nurses and approximately 200 patients in these five immunodeficiency centers. Each center was responsibility for recruiting patients to this study.

A sample of convenience was applied. The inclusion criteria for patients were that they had been diagnosed with PAD for at least six months and could understand and engage in a dialogue. The Swedish inclusion criteria for nurses were that they worked as immunodeficiency nurses during the pandemic. All staff were asked to participate. A total of 20 people with PAD were approached and 12 decided to participate. Out of 15 nurses, there were 12 that decided to participate. The reasons given for non-participation were personal reasons or that they did not have the opportunity to participate in the planned interviews. The characteristics of the participants are found in Table 1 and Table 2.

### 2.1. Data Collection

Patient participants were given a choice of five occasions to participate, and these five focus group interviews were conducted from January to April 2023. Each focus group was led by the first author (RF) or the last author (CP), who altered between the roles of facilitator and observer. To avoid the risk of bias, the interviews were conducted by a person with no care relationship with any of the participants. The facilitator led the interview following an interview guide. The observer took notes during the interview and concluded the discussions at the end of the interviews, asking if the participants wanted to clarify or add anything. Both researchers had previous experience in conducting focus groups (fg). The focus group interviews were mainly conducted digitally (n = 4), with one conducted in a hybrid setting (digital and on-site in a conference room at the hospital). Participants were informed about the objectives of the study and were obliged to respect the integrity and confidentiality of all group participants. They were also informed that they could decline to participate, and that the audio of the group discussion would be recorded and supplemented with the observer’s notes on the participants’ responses. The focus group interviews lasted from approximately 40 to 70 min, and each interview was transcribed verbatim. A semi-structured interview guide was created jointly by all authors, containing questions about how people with PAD experienced their life situation during the pandemic. Questions were asked about how they related to the public health authority’s recommendations and what support they sought during the pandemic. They were also asked to describe how they felt about the information they received on infection control, vaccinations and other precautions. For each question, probing questions were asked, for example, “can you explain more” or “did I summarize what you said correctly”.

### 2.2. Data Analysis

Inductive content analysis based on the description of Elo and Kyngäs, (2008) was performed [19]. The transcribed text in Swedish was read through several times to gain an understanding of the content. Subsequently, units of analysis were selected, which were one or more sentences with meanings related to the purpose of the study. During the organization phase, the authors (RF and CP) read the transcribed data several times to become familiar with all the data. This meant going back and forth between data and selecting sentences in the original text that corresponded to the study’s aim. If multiple meanings were selected, condensation was used to reduce the text. In the next step, the meanings and sentences were given a code, still close to the original text. Until this step, patient and nurse interviews were handled separately in Excel spreadsheets. In the open coding phase, both patient and nurse files were merged into one Excel spreadsheet and the codes were divided into generic categories and subcategories. The analysis resulted in one main category, two generic categories and six subcategories. To achieve credibility in the analysis, two authors were responsible for the analysis (RF and CP) and after a preliminary analysis was performed, the other authors were involved to discuss preliminary results, before adding their perspectives, until all authors reached consensus on the results, which was carried out in three different rounds [19]. An example of a coding tree is described in Table 3. The Consolidated Criteria for Reporting Qualitative Research (COREQ) checklist has been used as the basis of reporting for this study, and a detailed description can be found in Appendix B [20].

The study was approved by the Regional Ethical Review Board in Lund, Sweden (EPN 2019-04965). All procedures were performed in accordance with the Declaration of Helsinki, where the ethical principles of autonomy, benefit, non-benefit and fairness were considered. Written and oral information about the study was provided to each participant before they gave their informed consent to participate in the current study.

## 3. Results

“From crisis to a new orientation” describes the main category, including both people with PAD and nurses. The main category is built on two generic categories as follows: consequences for managing everyday life and both obtaining and offering support from the healthcare service (Table 4). “From crisis to a new orientation” describes the journey from uncertainty, worry and sometimes fear of the pandemic’s effects on those living with PAD. In the beginning, several factors from the outside world affected these feelings and influenced the managing of everyday life. This was illustrated by the rapid day-to-day changes when new recommendations were revealed. As the pandemic continued, more knowledge was developed, which helped with coping with the situation better. To be able to handle this uncertain situation, both people with PAD and nurses described the importance of receiving and offering specific support from health services. There was uncertainty about whether general information was sufficient, given that they belonged to a risk group. When the participants reflected on the time of the pandemic, they described how chaotic and uncertain the situation felt in the beginning, but as time went on, and as the transition to a new orientation took place, they adapted to the situation caused by the COVID-19 pandemic.

### 3.1. Consequences for Managing Everyday Life

This generic category consists of four subcategories as follows: attitudes from others, adaptations to different situations, social isolation, and being concerned about the emerging situation. Concerning “attitudes of others”, participants fully agreed that other people in the community suddenly had a better understanding of what it was like to deal with the potential risks of contracting a potentially severe infectious disease. People with PAD also described how others in society showed more respect, but they had also experienced lack of respect, especially in terms of adhering to the recommendations of social distancing and hygiene procedures. The pandemic led to a positive effect for people with PAD—it was easier to describe to other people what it is like to live with the fear of contracting an infectious disease.


*“It has been easier to explain, I think, to those around me, but this is how it always is for me.”*

*(fg 1—person with PAD)*


There were some differences between participants in how they described they were “adapting to a new situation”. According to people with PAD, they had taken this opportunity to talk to others about how to relate to the recommendations of the PHAS. They agreed that they had confidence in what the PHAS described in the media, but they also expressed that different opinions from experts and variations in recommendations issued by the local county medical officers created concerns and confusion. This was particularly evident when it came to wearing face masks. Nurses described that they rarely recommended face masks and that many people had difficulties with using face masks correctly. According to the nurses, some people with PAD felt safer and could accept higher risks when wearing face masks. Nurses closely followed new recommendations and adjustments as the pandemic continued. Some people with PAD searched for information on the internet, while others contacted their immunodeficiency center at the local department of infectious diseases to ask questions. Another issue raised by participants was that traveling to work could be considered a risk and was associated with the fear of becoming infected. According to the nurses, several people with PAD wanted a certificate stating that they could not work due to their PAD diagnosis. They also mentioned that there were people with PAD who wanted their children to stay home from school to minimize the risk of infection. There was a lack of information transfer regarding how their PAD condition could be affected by a potential infection and what vaccination effect they could expect.


*“It took quite a long time to find out if I belonged to a risk group and therefore being allowed to receive vaccination early and before others.”*

*(fg 1—person with PAD)*


During the pandemic, people with PAD experienced more days without symptoms of infection (such as coughing, cold symptoms, and fever), which was considered a positive effect of the pandemic related to “social isolation”. Furthermore, they expressed that they were socially isolated even before the pandemic, especially during the winter months, to reduce the risk of respiratory infections. Social isolation suddenly became a normal situation for both the public and people with PAD, and they became more socially isolated. The difference for people with PAD during the pandemic was that they felt even more isolated because all citizens in the society were recommended to isolate themselves due to the risk of being infected or infecting others. People with PAD expressed concerns regarding being isolated during the pandemic, elaborating on the feeling of loneliness and how they had to withdraw from social events. Nurses confirmed these statements from the patients, as they received many questions about isolation and described that there were differences in how people with PAD experienced social isolation during the pandemic. According to the nurses, some patients took social isolation to an extreme level, which was often associated with a fear of meeting other people in real life due to the risk of contracting COVID-19.


*“My patients—many of them—thought that if they contracted COVID-19 they would die, so they were really scared and started to isolate themselves completely.”*

*(fg 2—nurse)*


The pandemic led to several consequences that influenced the ways of managing everyday life, which are illustrated in the subcategory “being concerned about the emerging situation”. Due to the pandemic, people with PAD tried to meet family members outdoors and were careful to maintain physical distance. Some decided to stop using public transport in fear of being too close to potentially infected people. Those who did not habitually use web-based forums, such as online video conferences, forced themselves to learn, since it was the only way they could maintain contact with grandchildren and friends. Given the risk of transmission, people with PAD described how they chose to refrain from participating in social activities, even if it made them feel mentally worse. However, some discovered new leisure activities, such as gardening. The most important consequence of the pandemic was the persistent fear of being infected and the lack of knowledge on how this would affect them due to their condition. Consequences described by the nurses included the difficulties in responding to patients’ concerns and fears related to the lack of clinical experience and evidence.


*“I think there was a bit of a lack of information actually; I would have liked to have more information from the start.”*

*(fg 2—person with PAD)*


Other consequences for people with PAD were that they had difficulties booking appointments for bacterial respiratory infections in primary care, resulting in a diagnostic delay. The nurses said that most people with PAD seemed to be healthier during and after the pandemic, which was likely related to social isolation. However, the positive effects on physical health did not outweigh the negative effects of social isolation on mental health. The nurses found that people with PAD were afraid of all potential risks of being infected by COVID-19. Even when restrictions were lifted by the PHAS in early February 2022, people with PAD continued to choose to keep their physical distance from others and to wear face masks.


*“It was quite a long period that we did not have any answers; the first year we did not have any knowledge about what could happen to patients with PAD and which ones could be at risk of becoming severely ill.”*

*(fg 1—nurse)*


### 3.2. Obtaining and Offering Support from Healthcare Services

The last generic category consists of the following two subcategories: “back up” and “accessibility”. There was less understanding about the situation for people with PAD during the pandemic among healthcare providers outside of immunodeficiency centers. These experiences shed light on the need for back up within the healthcare system. When turning to primary care providers, people with PAD supported that they were sometimes questioned and not provided with adequate care but received better support when turning to the immunodeficiency center and their nurses. Respondents felt that a written care plan describing their specific problems and information about their specific, rare PAD condition could improve support from primary care. People with PAD also expressed concerns about the limited knowledge in the early stages of the pandemic, as well as the uncertainty about whether COVID-19 vaccinations had any effect on their condition. This uncertainty was associated with worry and anxiety. In general, they states that the support from the immunodeficiency centers as outstanding and overall found the care to be of high quality. The nurses also described the issues surrounding vaccinations, a topic that patients had many questions about, and which they, as nurses, had difficulty in answering due to lack of knowledge. This changed after a while, when more scientific studies on COVID-19 vaccine responses were published. Nurses expressed that they wanted to give their best support to the patients to make them feel safe, and to accomplish this, teamwork was needed.


*“I am all alone so it is hard when they reject you…next time I make contact I may have made contact ten times without being allowed to come [to primary care].”*

*(fg 3—person with PAD)*


“Accessibility” was considered necessary for receiving (patients) and offering (nurses) support during the pandemic. People with PAD described that they were invited to meet digitally instead of physically visiting the hospital, which was important from an accessibility perspective. They also saw a risk of becoming a burden to the healthcare system, if they were advised to go to the emergency department and were stuck there for hours, when they should have been referred to a health center instead. The nurses confirmed the perception of people with PAD that it was important to have access to online visits. The nurses also felt that they had more time to counsel people with PAD about self-care information, as other outpatient services were greatly reduced. The nurses were proud that they were able to launch telephone visits and counseling at a short notice at the beginning of the pandemic. This change ensured accessibility for people with PAD, without increasing the risk of contracting COVID-19.


*“Then we also got started quite quickly with video visits.”*

*(fg 2—nurse)*


## 4. Discussion

Our main findings show that the pandemic affected the daily life of people with PAD in different ways, which was partly through the lack of specific recommendations for them and partly through the uncertainty regarding the need for extreme social isolation. This study also illustrates how support from healthcare providers was quickly transformed into tele-healthcare encounters to ensure accessibility. Another finding is the need to clarify ways to better coordinate primary and specialized care.

The respondents in this study, as well as the general population, largely followed the PHAS recommendations, indicating a high level of trust in both the healthcare system and the authorities [21]. According to Sowers and Galantino, people with severe immunodeficiency reported a high degree of concern that they themselves or someone close to them would become seriously ill [22], a finding which also emerged from this study. The nurses faced a challenge with regard to the advice to use a face mask, which was a new precaution for respondents, related to the fact that the recommendation was unclear [2] and that the PHAS found that there was a lack of evidence for this [23].

According to our results, it was emotionally difficult for patients to distance themselves from grandchildren, and this has been demonstrated before, in the ways distance affects people’s physical and mental health [24]. This means that healthcare providers can learn several lessons from the pandemic. When counseling, it is important that the healthcare provider builds trust and understands the patient’s attitudes to minimize the impact on mental health. It is important to understand individuals’ trust and attitudes when counseling or giving support [25]. Nurses and other healthcare professionals should focus on empowering the patient regarding health-related issues through a holistic approach [26]. People with PAD demonstrated increased anxiety and fear during the COVID-19 pandemic, which was most pronounced when contacting unfamiliar people or visiting public places [27]. Therefore, it is of particular importance to focus on psychological well-being when worry and anxiety rise in people with chronic diseases during a public health crisis [22].

In this study, those living with PAD felt healthier, reporting a decreased burden of infections because of social isolation, which they considered positive. Significant reductions in transmissible diseases during the pandemic have been reported across the world [28]. Participants even expressed that they were largely infection-free during the pandemic years. However, the downside of the reduction in infections was an increased feeling of loneliness and a lack of social interaction. The negative effects on psychological health due to the need for social isolation to slow down the transmission of SARS-CoV-2 have also been described by others [29]. Preventive measures, such as social isolation and disconnection from everyday context, increase the risk of poor mental health, which is an important aspect to consider [30].

Both people with PAD and nurses highlighted how rapidly healthcare providers made the transition from physical encounters to tele-healthcare, during the early phase of the pandemic, to respond to and provide support related to concerns and anxiety about contracting COVID-19 [31,32]. This shows that tele-health is an important step towards enabling continuity of care, especially in the event of a public health crisis [31]. However, there may be challenges associated with changing counseling from physical meetings to online support [32]. Nevertheless, online support may reduce patient travel, ease congested clinics and be positive for people with PAD, as they are vulnerable and immunologically impaired [31]. This means that the pandemic had positive effects by accelerating the introduction of tele-healthcare encounters and support from healthcare staff using online meetings [33]. This new way of facilitating care can benefit both healthcare providers and patients, and it builds a foundation for the further development of future healthcare in terms of the use digital tools.

The evidence suggests the importance of an individual care plan. Patients in Sweden have described poorer coordination of their care between different centers of the healthcare system, compared to patients in ten other European countries [34]. One explanation for the poor care coordination in Sweden may be that just over 60% of the population has a primary care physician for regular contact, compared to 90% in most of the other countries. The use of individual care plans for people with chronic diseases has been reported to be associated with more patient-centered care [35]. Person-centered care is based on patient needs and participation, and shared decision making can improve adherence to treatment and other recommendations [36]. To summarize, the development and implementation of individual care plans may improve the care of people with PAD due to the improved collaboration between primary healthcare and immunodeficiency-specialized care.

### Strengths and Limitations

The trustworthiness of the study is described according to Lincoln and Guba (1985) [37]. To establish credibility, concerning the confidence of the findings, we approached both people with PAD and nurses, which is a strength. A weakness is that we do not know anything about the perspectives of those that declined to participate in the study. To show that the findings are applicable, we approached different sites to expand the possibilities for different perspectives, due to concerns with local differences in how care is delivered. A potential weakness to the transferability is that care may differ between countries, and therefore, the results may only be applicable in a Swedish context. Dependability was considered by using an audit trail, showing examples of how the raw data were transformed into codes, subcategories and categories. Finally, to establish confirmability, the extent to which the findings are shaped by participants and not researcher bias, we used the authors’ different approaches in analysis discussions (reflexivity). Three of the authors (RF, ÅN, and SN) have deep knowledge and preunderstanding of the area of study, whereas two of the other authors (BÅ and CP) are outside the field of primary immunodeficiency. In total, three separate sessions for data analysis discussion among all authors were conducted, and an analysis emerged after each session. To illustrate the connection to the participants’ statements, quotations illustrating each subcategory were used, strengthening the study’s results.

## 5. Conclusions

The burden of disease is ever-present for people with PAD, and anxiety and fear rise in new circumstances, such as public health challenges. The COVID-19 pandemic has helped us understand how respiratory infections can be drastically reduced if people with symptoms of respiratory symptoms adhere to social isolation and other precautions. Telemedicine encounters are useful tools for facilitating healthcare visits when the risk of the transmission of infections is increased. Individual care plans developed by patients, together with different healthcare providers, may increase the possibility of integrated care. This can also be applied to other chronic conditions as well. Still, it is important to continue researching ways to improve care and decrease the burden of treatment for people with PADs, to provide the support that is needed. Based on the study’s results, our recommendation is to develop telemedicine meetings and facilitate information transfer between primary and specialist care, for example, through individual care plans. Close collaboration between the Swedish Medical Association for Primary Immunodeficiency, the specialist clinics and the Primary Immunodeficiency Organization can be supportive in future pandemics. This would make it easier to direct targeted information, given through websites and webinars, to this patient group.

## Figures and Tables

**Table 1 nursrep-15-00104-t001:** Sample characteristics of people with PAD in the current study.

Total Number	Age Years Median (Range)	Gender	Cohabitant/Single	Employment	COVID-19	Vaccine Uptake
12	59 (28–83)	7 men 5 women	10 cohabitant 2 single	6 working 1 disability 5 retired	7 confirmed (1 of which in patient care) 5 no symtoms	11

**Table 2 nursrep-15-00104-t002:** Nurse charecteristics in the current study.

Total Number	Age Years Median (Range)	Gender	Years of Experience (Range)
12	57 (43–68)	Women only	14 (1–25)

**Table 3 nursrep-15-00104-t003:** Example of coding tree.

Meaningful Unit	Condensation	Subcategories	Generic Categories	Main Category
“I look back on the pandemic then i felt good purely so, then i thought it was hard mentally to sit at home and not see anyone—but just for the immunodeficiency it went great”	Infection-free but mentally hard to sit at home	Social isolations	Consequences for managing everyday life	From crisis to new orientation

**Table 4 nursrep-15-00104-t004:** Description of the main category, the two generic categoris and the subcategoris that emerged in the analysis.

Main Category	Generic Category	Subcategories
From crisis to new orientation	Consequences for managing everyday life	Attitudes from others Adaptations to different situations Social isolation Being concerned about the emeerging situation
	Obtaining and offering support from the healthcare service	Back up Accessibility

## Data Availability

The data that support the findings of this study can be made available from the corresponding author (R.F.) upon reasonable request.

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
