# Peer review of "Experience of the COVID-19 Pandemic in the Care of Patients with Predominantly Antibody Deficiencies (PADs)—A Qualitative Study with Perspectives from Both Patients and Nurses"

_nursrep, 2025, doi:10.3390/nursrep15030104_

Round 1

Reviewer 1 Report

Comments and Suggestions for Authors

this study is a phenomenological focus-group study that sought to understand how high-risk patients, those diagnosed with PAD contended with social isolation. The authors collected data from patients and nurses via multiple focus-groups conducted at five occasions with each focus group led by one of the study authors. Data analysis was conducted by two of the authors by using a coding tree. 

The focus group design is appropriate for this type of phenomenological study. No modifications needed.

The data analysis is appropriate for this type of study. No modifications needed.

The results reached by the authors are responsible and logically supported by the transcript excerpts reported in the study manuscript. No modifications needed.

The conclusions reached by the authors are appropriate given the data and themes reported in the analysis section of the manuscript. One suggestion -- add narrative to the conclusion that shows how the findings inform healthcare providers how to approach similar concerns in a future pandemic or localized environmental catastrophe that impacts patients with PAD or similar concerns. This helps generalize the value of the study to a broader set of future problems

Author Response

Thank you very much for taking time to review this manuscript. Please find the detailed responses below and the corresponding revisions/corrections highlighted/in track changes in the re-submitted file.

Comments 1: The conclusions reached by the authors are appropriate given the data and themes reported in the analysis section of the manuscript. One suggestion -- add narrative to the conclusion that shows how the findings inform healthcare providers how to approach similar concerns in a future pandemic or localized environmental catastrophe that impacts patients with PAD or similar concerns. This helps generalize the value of the study to a broader set of future problems

Response 1: Thank you for pointing this out. We agree with this comment. Therefore, we have added a narrative in the conclusion.

Conclusion 426-430

Close collaboration between the Swedish Medical Association for Primary Immunodeficiency, the specialist clinics and the Primary Immunodeficiency Organization can be supportive in future pandemics. This would make it easier for targeted information given through websites and webinars directed to this patient group.

Reviewer 2 Report

Comments and Suggestions for Authors

Thank you for your paper, and I enjoyed reading it. I wish you all every success in your ongoing research.

The introduction was clear and set the context for the study.

Table 1: I am not sure what you mean by ‘unrecognized’ in the COVID-19 column -please clarify, perhaps by including a legend.

Table 1: There were 12 participants, but the different columns do not all add up to 12, so clarification is needed here too.

Data collection and analysis: Well done on aligning your work to the COREQ guidelines. A couple of additional details would help strengthen these sections.

What was the relationship (if any) between the authors and the participants? This is to determine whether or not the power relationship was considered.

Did the authors undertake a reflexivity exercise, and if so, it would be useful to highlight decisions made based on this.

Were the focus groups conducted in English? If not, what language, and how was translation managed? Were the transcripts verified, and if so, by whom?

Results: 3.1. These findings were clearly presented. You mentioned that both physical and mental health were impacted by COVID-19. There was some elaboration on the physical issues, and I am wondering if you could expand a little more on the mental. You refer to social isolation. Could you give a little more detail here?

Line 316: what is context for rejection? I am not sure how this was interpreted in relation to the preceding statements. Would a different quote illustrate your point more effectively?

Discussion: paragraph 348 – 358. There are a number of different concepts in this section: ‘emotionally difficult to distance themselves’ – and I don’t know how this links with ‘understand individuals’ trust and attitudes’, or empowering patients. Please consider re-framing the paragraph for clarity.

Minor editing:

Line 257: re-phrase for clarity

Line 276: ‘even if they lacked this’ I am not sure what you mean here.

Author Response

Thank you very much for taking time to review this manuscript. Please find the detailed responses below and the corresponding revisons/corrections highlighted/in track changes in the re-submitted files.

Comments 1: Table 1: I am not sure what you mean by ‘unrecognized’ in the COVID-19 column -please clarify, perhaps by including a legend.

Response 1: Thank you for pointing this out. We have clarified this in Table 1.

Comments 2:  Table 1: There were 12 participants, but the different columns do not all add up to 12, so clarification is needed here too. 

Resonse 2: Thank you for pointing this out. The Table 1 is updated.

Comments 3: Data collection and analysis: Well done on aligning your work to the COREQ guidelines. A couple of additional details would help strengthen these sections.

What was the relationship (if any) between the authors and the participants? This is to determine whether or not the power relationship was considered.

Response 3:  Thank you for the comment. To avoid the risk of bias, the interviews were primarily conducted by a person who did not have a patient relationship see line…???

Comments 4: Did the authors undertake a reflexivity exercise, and if so, it would be useful to highlight decisions made based on this?

Response 4:  Thank you for the comment.  A reflexivity approach was made by taking advantage of the different perspectives each author has. Three of the authors (RF, ÅN, SN) have a preunderstanding about the area of study, whereas two of the authors (BÅ, CP) are outside the field of primary immunodeficiency. This was given us an opportunity to weight the discussions during the analytic procedure. This is illustrated in the method section, 4.1.

Comments 5: Were the focus groups conducted in English? If not, what language, and how was translation managed?

Response 5:  Thank you for the comment.  The content analysis was conducted in Swedish on the basis that all interviews in the focus groups was performed in Swedish. We did the content analysis in our native language and after the analysis was finished, the material was translated to English.

Comments 6: Were the transcripts verified, and if so, by whom?

Response 6:  Thank you for the comment.  The transcripts have been verified by the other co-authors and Magnus Burström on several occasions.

Comments 7: These findings were clearly presented. You mentioned that both physical and mental health were impacted by COVID-19. There was some elaboration on the physical issues, and I am wondering if you could expand a little more on the mental. You refer to social isolation. Could you give a little more detail here?

Response 7:  Thank you for the comment. We have reframed the paragraph for clarity. 

The mental health was affected, for example, by not be able too participate in regular activites such as meeting family members,  going to gym or theater related to fear of being infected by COVID-19.

Comments 8:  Line 316: what is context for rejection? I am not sure how this was interpreted in relation to the preceding statements. Would a different quote illustrate your point more effectively?

Response 8: Thanks for the comment. We have reframed the paragraph for clarity. 

“That they are confident that they can call and still get confirmation” fg 1 -nurse

Comments 9: Discussion: paragraph 348 – 358. There are a number of different concepts in this section: ‘emotionally difficult to distance themselves’ – and I don’t know how this links with ‘understand individuals’ trust and attitudes’, or empowering patients. Please consider re-framing the paragraph for clarity.

Response 9: Thank you for the comment. We have reframed the paragraph for clarity. 

 This means that healthcare providers can learn several lessons from the pandemic. When counselling, it is important that the healthcare provider builds trust and understands the patient's attitudes to minimize the impact on mental health. It is important to understand individuals’ trust and attitudes when counselling or giving support {25}. Nurses and other healthcare professionals should focus on empowering the patient regarding health-related issues based on a holistic approach {26}. People with PAD demonstrated increased anxiety and fear during the COVID-19 pandemic, which was most pronounced when contacting unfamiliar people or visiting public places {27}. Therefore, it is of particular importance to focus on psychological well-being when worry and anxiety increase in people with chronic diseases during a public health crisis

Comments 10: Line 257: re-phrase for clarity

Response 10: Thank you for the comment. We have reframed the paragraph for clarity. 

The difference for people with PAD during the pandemic was that they felt even more isolated because all citizens in the society was recommended to isolate themselves, due to the risk to being infected or infect others.

Comments 11: Line 276: ‘even if they lacked this’ I am not sure what you mean here.

Response 11: Thank you for the comment. We have reframed the paragraph for clarity. 

Given the risk of contracting COVID-19, people with PAD described actively choosing not to participate in social activities even if it made them feel mentally worse.

Reviewer 3 Report

Comments and Suggestions for Authors

The article addresses an important topic by focusing on the experiences of PAD patients and nurses during the COVID-19 pandemic, highlighting the unique challenges faced by this vulnerable group. The use of focus group interviews for data collection is appropriate, allowing for in-depth exploration of both patients and nurses personal experiences and perspectives. However, It would b valuable to hear how the patients´ and nurses´ data were combined.

The content analysis method identifies key themes such as the effects of social isolation and the shift to telemedicine. However, it does not state whether the content analysis method was deductive or inductive. This should be stated. The pattern of results is drawn as the analysis method was deductive, but I am not sure whether the authors of the article decided how to position the results based on the two different methodological choices. The results are clearly described, highlighting the importance of coordinated care and the role of specialist care in supporting PAD patients. The discussion provides valuable insights into the high level of trust in Swedish authorities and the benefits of telemedicine. Overall, the study highlights the need for better care.

The conclusion of the article highlights the critical role of specialist care in supporting PAD patients, especially during the pandemic. It highlights the importance of effective information transfer and coordinated care plans, suggesting that telemedicine can be a valuable complement to traditional care methods. The conclusion could be added by evaluation how the results of this study can be used also in the other patient groups than PAD patients care.

I would hope that the conclusions would provide some insight into how the results of this study can be applied to the treatment of other patient groups for whom respiratory and other seasonal infections are particularly harmful. Psychological well-being and its support as part of comprehensive patient care are key nursing skills and areas of expertise.

Author Response

Thank you very much for taking time to review this manuscript. Please find the detailed responses below and the corresponding revisons/corrections highlighted/in track changes in the re-submitted feels.

Comments 1: The article addresses an important topic by focusing on the experiences of PAD patients and nurses during the COVID-19 pandemic, highlighting the unique challenges faced by this vulnerable group. The use of focus group interviews for data collection is appropriate, allowing for in-depth exploration of both patients and nurses personal experiences and perspectives. However, It would b valuable to hear how the patients´ and nurses´ data were combined.

Response 1:  Thank you for pointing this out. In the text, we have added a clarification when the data sets were merged. Until this step, patient and nurse interviews were handled separately in Excel spreadsheets. In the open coding phase, both patient and nurse files were merged into one Excel spreadsheet and the codes were divided into generic categories and subcategories

Comments 2: The content analysis method identifies key themes such as the effects of social isolation and the shift to telemedicine. However, it does not state whether the content analysis method was deductive or inductive. This should be stated. The pattern of results is drawn as the analysis method was deductive, but I am not sure whether the authors of the article decided how to position the results based on the two different methodological choices. The results are clearly described, highlighting the importance of coordinated care and the role of specialist care in supporting PAD patients. The discussion provides valuable insights into the high level of trust in Swedish authorities and the benefits of telemedicine. Overall, the study highlights the need for better care.

Response 2: Thanks for the comment.The content analysis was inductive, which is added in the method section. 

Comments 3: The conclusion of the article highlights the critical role of specialist care in supporting PAD patients, especially during the pandemic. It highlights the importance of effective information transfer and coordinated care plans, suggesting that telemedicine can be a valuable complement to traditional care methods. The conclusion could be added by evaluation how the results of this study can be used also in the other patient groups than PAD patients care

Response 3: Great point, we have added that into the conclusion now. 

Comments 4: I would hope that the conclusions would provide some insight into how the results of this study can be applied to the treatment of other patient groups for whom respiratory and other seasonal infections are particularly harmful. Psychological well-being and its support as part of comprehensive patient care are key nursing skills and areas of expertise.

Response 4: Thank you, we totally agree, and our experience is that involving a counsellor in the team around a person with PAD is an advantage.